# When Super-Resolution Localization Microscopy Meets Carbon Nanotubes

**DOI:** 10.3390/nano12091433

**Published:** 2022-04-22

**Authors:** Somen Nandi, Karen Caicedo, Laurent Cognet

**Affiliations:** 1Laboratoire Photonique Numérique et Nanosciences, Université de Bordeaux, UMR 5298, 33400 Talence, France; nandi.somen@institutoptique.fr (S.N.); karen.caicedo-santamaria@institutoptique.fr (K.C.); 2Institut d’Optique and CNRS, LP2N UMR 5298, 33400 Talence, France

**Keywords:** single-walled carbon nanotubes (SWCNTs), super-resolution microscopy (SRM), nanoscience, nanoscopy, bionanotechnology, single-molecule study, single-particle tracking (SPT), near-infrared (NIR) probes

## Abstract

We recently assisted in a revolution in the realm of fluorescence microscopy triggered by the advent of super-resolution techniques that surpass the classic diffraction limit barrier. By providing optical images with nanometer resolution in the far field, super-resolution microscopy (SRM) is currently accelerating our understanding of the molecular organization of bio-specimens, bridging the gap between cellular observations and molecular structural knowledge, which was previously only accessible using electron microscopy. SRM mainly finds its roots in progress made in the control and manipulation of the optical properties of (single) fluorescent molecules. The flourishing development of novel fluorescent nanostructures has recently opened the possibility of associating super-resolution imaging strategies with nanomaterials’ design and applications. In this review article, we discuss some of the recent developments in the field of super-resolution imaging explicitly based on the use of nanomaterials. As an archetypal class of fluorescent nanomaterial, we mainly focus on single-walled carbon nanotubes (SWCNTs), which are photoluminescent emitters at near-infrared (NIR) wavelengths bearing great interest for biological imaging and for information optical transmission. Whether for fundamental applications in nanomaterial science or in biology, we show how super-resolution techniques can be applied to create nanoscale images “*in*”, “*of*” and “*with*” SWCNTs.

## 1. Introduction/Background

Understanding the precise structural features and dynamical behavior of nano-architectures is an urgent need in many areas of modern science. As optical imaging techniques enable contactless and remote observations of specimens in their native environments, including living samples, much effort in recent decades has been dedicated to pushing the resolution of optical microscopes down to nanometer scales. This effort has been recompensed by the advent of so-called “super-resolution microscopy” (SRM) techniques [1,2,3], which are now revolutionizing the fields of physics, chemistry, biology and the bio-medical sciences. The bottleneck of optical microscopy has long been the problem of going beyond the diffraction limit (Abbe’s criteria) of optical microscopes and producing images of observed specimens with dimensions below ~200 nm (i.e., ~λ_em_/2; with the emission wavelength of λ_em_), down to the nanometer scale. SRM essentially uses the photocontrolled emission of fluorescent molecules to generate images of objects at the molecular level [4]. Several excellent reviews have been dedicated to SRM techniques and their applications, especially in biology, and we recommend the reader consult these reviews for further information [5,6,7,8,9].

The design, development and application of SRM approaches dedicated to the use or study of synthetic nanomaterials is, however, a more recent field of research and has shown great promise for several types of nanomaterials, such as spherical nanoparticles (e.g., gold or silver nanoparticles, etc.), 2-D hexagonal boron nitride (h-BN), supramolecular polymer-based nano-assembly, gold nanorods [7,10,11,12,13] and single-walled carbon nanotubes (SWCNTs). For many nanomaterials, however, SRM is challenging due to the limited PL brightness and high photobleaching properties of the nano-objects, as well as the difficulty of labeling them with fluorophores. In contrast, SWCNTs display rich optical (no photobleaching or self-blinking), physical and chemical potentialities together with unique sensing capacities. They are also versatile near-infrared (NIR) emitters covering the biological/telecom windows. As a consequence, the characterization of SWCNTs with super-resolution techniques in terms of exciton localization, quenching or luminescent defects, and more generally photophysical properties, has emerged and triggered strong interest due to the implications for many applications in materials science, nanoelectronics, photonics, quantum information and biophotonics, for example. There are ancillary techniques such as atomic force microscopy (AFM) that can be used to measure, e.g., the length of nanotubes, which is one of the important physical parameters, but they are not suitable for studying the optical properties of nanotubes. Additionally, it is important to note that most of the super-resolution microscopy approaches applied to other nanomaterials mentioned above were typically constrained to the study of some of the internal structures of these nanostructures, while SWCNTs have shown the supplementary ability to generate super-resolution data for complex environments and in the NIR regime. This makes them particularly attractive for high-resolution bio-imaging.

In this review, we therefore highlight SWCNTs as a particular class of nanostructure in the context of SRM. More specifically, we provide an overview on pioneering work where SRM is developed to provide nanoscale images “*in*”, “*of*” and “*with*” SWCNTs (Figure 1).

We have deliberately restricted this review to fluorescence microscopy. However, we would like to mention that the development of label-free super-resolution optical methods for studying nanostructures, including SWCNTs, is a fascinating endeavor. In a very recent study, Wang et al. [14] developed a label-free super-resolution imaging approach called structured illumination Raman microscopy for the characterization of several nanomaterials including SWCNTs in situ. They demonstrated that they could achieve a lateral resolution of ~80 nm with this label-free imaging modality.

In the following, we first provide an overview of current super-resolution strategies, including their advantages and shortfalls, and then discuss recent advances and developments in characterization techniques involving SWCNTs and their applicability for investigating more complex and heterogeneous biological systems. Throughout the text, we mainly discuss and focus on the central role of super resolution in promoting SWCNTs as useful nanomaterials to be explored rigorously.

## 2. Discussion

### 2.1. A Brief Overview of SRM Techniques

All the well-established SRM approaches can be schematically classified into two families. On the one hand, we have illumination pattern-based methods allowing the extraction of nanoscale structures in ensemble measurements, such as nonlinear structural illumination microscopy (SIM) [15], stimulated emission depletion (STED) [16] and reversible saturable/switchable optical linear fluorescence transitions (RESOLFT) [17] and related techniques. On the other hand, we have single-molecule detection and localization-based approaches, generically known as single-molecule localization microscopy (SMLM); for example, (direct) stochastic optical reconstruction microscopy ((d)STORM) [18,19], photoactivated localization microscopy (PALM) [20,21], points accumulation for imaging in nanoscale topography (PAINT) [22] and related techniques. These approaches are conducted by implementing microscopic setups in total internal reflection fluorescence (TIRF) mode, in widefield and/or in confocal systems. While illumination-based modalities do not rely on the use of specific fluorescent molecules (but bright ones), SMLM methods require stochastic on/off photodynamics (blinking events) for specific fluorophores, reflecting the limitations in the choice of probe molecule. Over time, the different techniques have reached a certain degree of maturity through the development of more advanced features that allow both high spatial and temporal resolution, such as saturated or nonlinear and instant SIM (spatial resolution ~50 nm) [23,24], 4Pi (axial resolution ~40 nm) [25], DNA-PAINT (spatial resolution ~5 nm) [26], light-sheet fluorescence [27] and minimal photon fluxes (MINFLUX) [28]. For instance, MINFLUX offers ~2 nm spatial resolution with microsecond (µs) temporal resolution. In addition, we should mention some of the analytical SRM techniques, such as super-resolution optical fluctuation imaging (SOFI) [29] and super-resolution radial fluctuations (SRRF) [30], which can be interesting in certain applications due to their instrumental simplicity, although they are not capable of reaching the same resolution as mentioned above. However, all these SRM techniques today share one limitation, which is that they use fluorescent molecules (common organic labels) emitting in the visible and far-red ranges (400–750 nm) of the electromagnetic spectrum. For instance, SMLM has rapidly expanded the super-resolution domain thanks to the design and development of a variety of photoswitchable emitters, all in the visible or far-red region. This limitation attracted a lot of attention in the scientific community and encouraged the identification and development of NIR emitters [31,32]. In this context, SWCNT is an appealing candidate as one of the most emblematic classes of NIR nanoprobes. This is confirmed by the growing number of reports based on their use in conjunction with SRM.

### 2.2. SWCNTs: Structure and Photophysical Properties

#### 2.2.1. Pristine SWCNTs

Carbon nanotubes (CNTs) are quasi one-dimensional (1D) nanostructures, rolled up in a graphene lattice of sp^2^ hybridized carbon atoms to form a hollow cylindrical structure with a very high length-to-diameter aspect ratio (0.4 to 10 nm diameters with lengths of several hundreds of nm to µm). When the tubes consist of a single layer of graphene, they are called single-walled carbon nanotubes (SWCNTs) and hold unique physical, chemical and opto-electrical properties compared to double or multi-walled carbon nanotubes. Since their discovery in 1991 [33], they have been extensively used for both fundamental studies and from applicative perspectives. The discovery of their NIR photoluminescence (PL) properties (λ_em_ ~870–2500 nm) [34], which falls in the so-called “*tissue transparency window*” (because of less absorption, reduced scattering and better penetration of light) and “*telecom windows*” (1300–1500 nm), soon laid the foundation for using SWCNTs in bio-imaging, sensing and quantum source applications [35,36,37].

SWCNTs are, in principle, defined by the two structural indices (n,m) referring to the diameter and chiral angle of the nanotubes, where n and m are integers. This chiral index is useful for determining the electronic structure of CNTs, including their metallic or semiconducting character. For instance, when |m − n| = 3*k*, the SWCNT is metallic, but if |m − n| = 3*k* ± 1, it is semiconducting in nature (where *k* is an integer). The optical properties of these nanoparticles can be determined from electronic transitions within the 1D density of states (DOS), as shown in Figure 2a. Upon excitation with a suitable light at their optical transition energies, photon absorptions lead to the creation of excitons (Figure 2), which are basically defined as electrostatically bound electron–hole pair systems, with a typical size of ~2 nm for semiconducting (6,5) SWCNTs [38]. Photoexcited excitons are mobile along the nanotube axis, undergoing excursion around ≥100 nm, often called the exciton diffusion length [39,40]. When an exciton comes to the ground state after radiative recombination, a photon is emitted, leading to luminescence in the NIR window (λ_em_ > 870 nm to 2500 nm [34,41]) with emission energy depending on the nanotube chirality [42]. For example, the first-order (E_11_) and second-order (E_22_) excitonic energy resonances for (6,5) SWCNTs are at ~985 nm and ~565 nm, respectively; for (7,5) SWCNTs they are at ~1025 nm and ~647 nm, respectively; and for (10,2) SWCNTs they are at ~1060 nm and ~737 nm, respectively [42,43] (see Figure 2b).

Photoluminescence quantum yields (PLQYs) for pristine SWCNTs are known to be extremely low (<1%) [44,45]. This is mainly due to the combination of two effects. On the fundamental side, it is due to the presence of a low-lying “dark” state sitting just ~5–100 meV below the optically allowed E_11_ bright exciton, from which the energy is thermally lost [46], but also due to the nonradiative decay of the localized exciton involving multiphonon decay and a phonon-assisted indirect exciton ionization mechanism [47]. In addition, extrinsic effects must also be taken into consideration. SWCNT PL is indeed extremely sensitive to local chemical defects in the nanotube structure but also to chemical or dielectric environments [39,48], which can all together lead to quenching and reduced PLQYs. The brightness of individual SWCNTs is thus highly variable and depends on the structural perfectness and the local environment, leading to reported values of PLQYs ranging from a few percent down to 0.01%. As it is discussed below, SWCNT PL sensitivity to local chemical environments can, however, be advantageously used in the context of high-resolution imaging; other applications include the development of chemical sensors, for instance. It is noteworthy that the roles of different surface-coating agents (or surfactants) in the aqueous dispersion of SWCNTs cannot be neglected with regard to aspects of photophysical properties, including brightness [49].

#### 2.2.2. Sp^3^ Defect-Functionalized SWCNTs

Shaping SWCNT PL through the implantation of sp^3^ defects into the sp^2^ carbon atoms situated on the nanotube surface came to the limelight as a promising way to improve the brightness of nanotube solutions [50,51,52,53]. The origin of this brightening lies in the possibility of trapping excitons at a specific location on the nanotube backbones, thus reducing the probability that excitons encounter quenching imperfections (whether structural or environmental) before their radiative recombination. Covalent sp^3^ functionalization of SWCNT is accompanied with a red-shifted E_11_* emission (by ~130–250 meV) from trapped excitons compared to the nanotube’s original excitonic E_11_ transition in the pristine SWCNTs [50,51].

Such sp^3^ functionalization of SWCNT is necessary to produce bright fluorescent ultrashort carbon nanotubes (usCNTs) with lengths below 100 nm. These usCNTs had long been sought for several applications, including for biological imaging, considering the high penetration depth of the light into the biological tissue, matching the SWCNT emission range. In this context, shortening the length of the SWCNTs could be seen as an advantage for mimicking bio-molecular dimensions. Unfortunately, the intrinsic NIR PL is known to be quenched in usCNTs because of their small size compared to the exciton diffusion length (<100 nm); specifically, nanotube ends are indeed efficient quenching sites [54,55]. Using sp^3^ defects, excitons can be prevented from reaching nanotube ends by local trapping, hence resulting in bright fluorescent usCNTs [52]. This first demonstration used a synthetic method involving first a mechanical cutting approach to produce ultrashort nanotubes, followed by sp^3^ (alkyl) functionalization. After that, another approach was proposed, which first involves sp^3^ (aryl) functionalization, followed by a chemical oxidation method to cut the nanotubes into bright ultrashort pieces [53], giving bright E_11_* luminescence at ~1150 nm (NIR-II window).

Considering all the aforementioned advantages in terms of physical, structural and optical properties, SWCNTs have turned out to be unparalleled NIR emitters down to the single-molecule level, with applications in diverse areas from quantum optics, telecommunication and sensing to in vitro and in vivo bio-imaging. These diverse applications have fostered a natural rapprochement between two disciplines: carbon nanotube spectroscopy and super-resolution microscopy.

### 2.3. Super-Resolution Imaging “of” and “in” SWCNTs: Unveiling the Intra-Nanotube Features with Subwavelength Resolution

Implementation of super resolution in the NIR range (with the emission wavelength of λ_em_ ≥ 1 µm) would be extremely valuable, one key factor being that the diffraction limit of an NIR optical microscope is larger than at visible wavelengths when using an objective with an equivalent numerical aperture (NA). More precisely, the resolution limit increases linearly with the emission wavelength (λ_em_) (equal to 1.22 λ_em_/(2 NA) for diffraction limited microscopes). This translates to resolutions of ~450–800 nm (for λ_em_ ~1000–1600 nm) using high NA objectives in comparison to ~250–350 nm at visible or far-red wavelengths, which is the emission range of common fluorophores used in bio-imaging, including super-resolution imaging. As already mention above, SRM in the NIR window ultimately suffers from a major obstacle, which is the lack of suitable probes in this wavelength range.

An initial demonstration of single-molecule localization analysis of long (~micrometer length) pristine SWCNTs blinking in an acidic environment was presented as early as 2008 [13] and represents the early age of SMLM. This opened the avenue toward super-resolution microscopy based on fluorescent SWCNTs. In the following section, we present the state-of-the-art of this nascent field.

#### 2.3.1. Localization of Bright Excitons in Pristine and Sp^3^ Defect-Functionalized SWCNTs

As the diffraction limit is considerably larger compared to the exciton diffusion length (~100–200 nm [39,40]), it is not possible to precisely localize the bright exciton along the SWCNT axis using a conventional microscope. In this context, the development of an innovative super-resolution imaging method revealed the landscape of excitonic emission along the nanotube backbone at the nanoscale resolution [13]. The method is primarily based on the on/off blinking of nanotubes, induced by stochastic molecular reactions (e.g., protonation/deprotonation) at the nanotube sidewall. SWCNTs (suspended in sodium dodecylbenzenesulfonate) in acidic environments display intensity fluctuations in different segments of the nanotube owing to localized individual protonation/deprotonation (Figure 3a). The process involves the injection of a hole close to the protonation site in the nanotube π-system and subsequent PL quenching of the exciton before recombining radiatively [56]. The computed differential images between successive frames clearly show positive and negative diffraction-limited intermittency spots (i-spots). These i-spots are then fitted to Gaussians from where the i-spot locations are recovered from their centroids with subdiffraction accuracy (~40 nm). Positive and negative i-spot maps have been retrieved from the multiple reaction sites along the SWCNTs, revealing bright exciton locations and, from them, their shape, nanotube lengths and orientations have been revealed (Figure 3b–e). Interestingly, as the nanotube ends act as quenching sites, the nanotube lengths deduced from their i-spot maps are smaller than their actual length. The analysis method is also able to detect the permanent defects present along the nanotube axis. This work took the first steps towards super-resolution imaging in SWCNTs. Notably, the concept of this approach highly resembles that of the PAINT [22] (e.g., uPAINT [57], DNA-PAINT [58]) methods involving molecular collisions and also (d)STORM [18,19] (spontaneous chemically induced blinking), where the positions of photoswitchable molecules separated at nanometric resolution are extracted.

When considering sp^3^ defect-implanted SWCNTs, it has been suggested that the excitons are trapped in the defects sites in these systems [59,60]. However, the direct demonstration of such exciton localization at defect sites cannot be performed due to the lack of resolution in standard optical microscopy, as discussed earlier. This is equally applicable for fluorescent usCNTs, which are based on sp^3^ defect implantation. In such systems, the length determination of usCNTs had to either rely on AFM or indirect methods [54,55]. Interestingly, SRM imaging of usCNTs has been performed, resulting in E_11_* exciton (emitting at the defect sites) localizations at single-defect sites with a precision of less than 25 nm (Figure 4a,b) and complementing the nanotube lengths revealed with AFM [52,53]. More precisely, following the concept described in [13], super-resolution was achieved by analyzing blinking behavior (PL intensity fluctuations). This analysis first demonstrated that defects could be studied and localized onto SWCNTs at the single-defect level. The super-resolved images suggested that, on average, two defect sites are separated by ~45 ± 10 nm, which agrees well with the nanotube length estimated with AFM (Figure 4b). It was therefore demonstrated that the introduced sp^3^ defects were mostly located at the usCNT ends where the bright E_11_* excitons are trapped and emit efficiently (Figure 4a). Notably, in these systems, the different defect sites behave independently to produce bright PL. A later study by Wang et al. [61] reproduced this general strategy to differentiate the position of different types of sp^3^ defects covalently introduced onto a single, long SWCNT. For this, they not only enhanced localization precision down to 15 nm by improving the signal-to-noise ratio of the images but also coupled localization and spectral identification of individual defects. More precisely, the instrumentation improvement included the use of an indium gallium arsenide (InGaAs) detector, cooled down to −190 °C and working in a non-destructive camera readout mode. They also used a gold (Au)-coated substrate to enhance the efficiency of the collection of the emitted photons by the objective. With this super-resolved hyperspectral imaging approach working in shortwave infrared, they could distinguish different defect types along the CNT axis at the single-nanotube level and demonstrated that defect implantations are inhomogeneous (Figure 4a,c).

In another study by Voisin and co-workers [62], they developed an elegant approach based on SRM to reveal excitons trapping in pristine SWCNTs along the nanotube axis at low (cryogenic) temperatures. In fact, they employed a similar strategy as that developed by van Oijen et al. [63] in 1998, which constituted the first experimental demonstration of SMLM. In both methods, the authors relied on the fact that, at cryogenic temperatures, different localized emitters give distinctive, sharp PL lines in which the positions of these lines depend on the local environment. Using a narrow-wavelength window selection, individual emission sites can be isolated spectrally and subsequently super-localized (Figure 4d,e) by Gaussian fitting, as is done in SMLM. The localization precision was achieved at a level below ~20 nm, making it possible to resolve exciton trapping sites separated from each other by ~100 nm along the SWCNT backbone [62]. The exciton localization at cryogenic temperatures was not ascribed to structural defects but rather to local heterogeneities at the nanotube/polymer interface. The authors validated their experimental observations using numerical simulation studies and showed that a single nanotube can act as a source of well-resolved, independent, confined emitters, which are useful for the generation of quantum light.

#### 2.3.2. Super-Resolution Imaging of Individual SWCNTs Using Photocontrolled Luminescence Intermittency

As shown in the above section, SMLM of SWCNTs can be achieved using the spontaneous blinking properties of the emitter induced by molecular collisions (as in universal (u)PAINT methods) or by the local chemical environment (as in (d)STORM). Another type of SMLM approach is PALM, which is based on the photocontrollable blinking of a single molecule. In practice, the development of PALM was intimately linked to the advent of photoswitchable fluorescent emitters in the visible spectral range [20,21]. Recently, Godin et al. [64] generated the first milestone for the extension of PALM in the NIR through the generation of photoswitchable SWCNTs and provided a proof-of-concept for SMLM using such nanotubes. More precisely, they presented a hybrid nanomaterial made of (10,2) SWCNTs (E_11_ transition at ~1065 nm) covalently functionalized with spiropyran–merocyanine (SP-MC), used to control the emission of SWCNTs and making them photoswitchable. SP-MC molecules were covalently attached to the SWCNTs by means of a nitrene-based cycloaddition reaction [65], obtaining fully conjugated SP-SWCNT hybrids through unique functionalization (Figure 5). This covalent approach does not hinder the conjugation between the SP molecule and sp^2^ carbon in CNTs. The first signs of photoswitching behavior appeared when a solution of SP-SWCNTs was illuminated with an ultraviolet (UV) light. A loss of around 50% of the signal of the SP-SWCNTs at the ensemble level took place within seconds. This partial quenching (which was later assigned to nanotube blinking at the single-particle level) was believed to be caused by a nonradiative recombination of the exciton induced by a charge transfer from the MC to the SWCNT favored by the UV illumination. When the UV lamp was taken away, the PL was recovered after a few tens of seconds [64].

A single-nanotube study revealed that nanotubes in fact blink upon UV light application. A forward–backward nonlinear filtering technique [66] was applied to the temporal intensity profiles of single nanotubes to analyze the blinking dynamics and compare the experimental observations with a simulation, taking into account the exciton diffusion properties, nanotube lengths and SP-MC functionalization density. Blinking of single nanotubes could then explain the partial quenching of nanotube solutions, as observed previously.

Godin et al. [64] took advantage of the induced blinking to apply an SMLM strategy (akin to PALM) and generate super-resolved images of the nanotubes. Blinking sites were fitted by two-dimensional Gaussian distributions, and the centroid was extracted with sub-wavelength precision. The localization precision achieved was <22 nm (Figure 5b). Due to the blinking performance, different segments of SWCNTs could be distinguished that otherwise would not be possible to resolve. Figure 5c exemplifies two segments resolved ~320 nm apart from each other, which it would not be possible to achieve with the conventional far-field image. This result demonstrates that SWCNTs functionalized with SP-MC molecules constitute a promising class of photoswitchable nanoprobe for the implementation of SRM in the NIR regime. This special type of emitter could also have several advantages for quantum information applications such as information storage elements.

#### 2.3.3. Super-Resolution Radial Fluctuation (SRRF) Nanoscopy of SWCNTs

Another recently developed analytical approach, known as super-resolution radial fluctuation (SRRF), is also able to provide super-resolved images of SWCNTs acquired from standard microscope setups without the need for fluorophore detection and localization [67]. The working principle of SRRF involves acquiring a set of images (in the order of hundreds to a thousand frames) and then, for each image, magnifying each pixel into subpixels, and a value called the “radiality” is calculated. The radiality is correlated with the probability of each subpixel containing the center of the fluorophore and is based on the information given by the point-spread function of the setup. Every pixel therefore contains a ”radiality stack”. Finally, an SRRF image is produced by using temporal correlations within the radiality stack.

In a recent study, Bisker and co-workers [68] applied this analytical approach to super-localize individual SWCNTs. NIR images were acquired using epi-illumination or TIRF-illumination setups. Samples with varying densities and different lengths of SWCNTs were imaged and treated with SRRF analysis. For different SWCNTs, the full width at half maximum (FWHM) of their cross-sections was calculated and averaged before and after applying SRRF, showing an improvement by a factor of ~5 in the determination of the FWHM. Besides this, SRRF was also able to distinguish close SWCNTs from each other. The average FWHM of sets of short and long SWCNTs was calculated and compared (Figure 6). This study demonstrated that the SRRF algorithms can be successfully applied in the NIR domain and are able to provide super-resolved images of nanotubes without complicated setups.

### 2.4. Super Resolution “with” Carbon Nanotubes: Single-Particle Tracking (SPT) Provides a Nanoscale Map of Complex Architecture

It is well-established that information about the local structural architecture of complex environments can be better revealed at the nanoscale regime by single-particle tracking (SPT) [69,70] compared to other single-molecule methods such as fluorescence correlation spectroscopy [71,72] or fluorescence recovery after photobleaching [73], which offer only ensemble averaging information. In fact, SPT can be seen as a parent methodology of SMLM [74,75]. SPT primarily involves recording single-particle diffusion in a continuous way in long video frames, followed by super-localization analysis of the single emitter in each video frame and reconstruction of its trajectory with specialized algorithms [76,77]. The collection of localization points along trajectories can then be used to deduce dynamically super-resolved images of the local environment explored by the particles, as in SMLM. For instance, SPT has been used to examine the random diffusion of single dye molecules in order to reveal dynamic information from various processes and systems, such as in crystalline hosts [78], ordered mesoporous structures [79,80], catalytic conversions [81] and heterogeneous biological architectures [82,83].

In this context, the development of SPT using SWCNT-based NIR video microscopy has allowed novel discoveries in different systems, including isolated cells and thicker 3D biological tissues [82,83,84]. In the following section, we focus on the application of SWCNTs with SPT for the exploration of the brain extracellular space (ECS), for which combining SWCNT assets and a super-resolution methodology has been the key feature.

#### 2.4.1. Unveiling the Extracellular Space (ECS) Features of the Brain at the Nanoscale Using Single Carbon Nanotube Diffusion

The brain ECS, the space in between brain cells, is crucial yet its morphology and physiology are still poorly understood, although it occupies one fifth of the brain’s volume. Through the super-resolved tracking analysis of SWCNTs diffusing in the brain ECS, recent work [83,85,86] has revealed the topology and diffusion properties of the ECS in different brain conditions (Figure 7). SWCNTs diffusing in ECS were super-localized and trajectories containing large amounts of information reconstructed. SWCNT tracking exhibits three decisive advantages over other emitters for such applications: first, their NIR emission perfectly matches the transparency window of biological tissues, allowing biological tissue to be imaged at depth (several tens of micrometers, i.e., several cellular layers). This is a stringent requirement when the elucidation of tissue architectures is targeted. Second, the high brightness and perfect photostability of individual SWCNTs’ PL allows ultra-long trajectories to be recorded (tens of thousands of points imaged at video rate). Third, the 1D character of the nanotubes is a unique feature that allows them to enter tiny channels of complex structures [87] thanks to their nanometer diameters while spending enough time there to explore their environment extensively thanks to their lengths (100 nm to micrometer lengths). As a result, trajectories contain an extensive number of localizations in local environments, enough to reveal both dimensions and diffusion properties. Super-resolved maps can thus be computed by cumulating the nanotube localizations, as in SMLM methodologies (Figure 7c,d). Godin et al. [83] demonstrated this approach in acute rat brain slices with a subwavelength precision of ~50 nm. They could then estimate the local dimensions of the brain ECS by analyzing the “shape” of the local area explored by the SWCNTs. A refined, non-parametric method to analyze such local areas was later proposed by Paviolo et al. [85]. From the mean-square displacement analysis, the instantaneous diffusion coefficient (D_inst_) could also be calculated. Comparison of rat organotypic tissue with mouse acute slices led to the revelation of the highly heterogeneous and tortuous features of ECS at the nanoscale level.

In another work, Soria et al. [86] investigated brain ECS modifications in the context of α-synuclein-induced neurodegeneration. For this, dopaminergic neuronal loss was induced in the substantia nigra of adult mice through unilateral inoculation of Lewy body fractions derived from Parkinson’s patients [88]. SWCNTs were tracked in acute slices of Lewy body-inoculated mice with a subwavelength precision of around 50 nm. Local, instantaneous, super-resolved diffusivity maps of ECS were created for both healthy and degenerative substantia nigra displaying heterogeneity in the ECS, as confirmed by electron microscopy in fixed tissues. The quantitative analysis showed that the diffusion in healthy brains was slower than in Lewy body-inoculated pathology mice. This work demonstrated that hyaluronan plays the central role for the variations in the ECS microenvironment and can have a severe impact on brain pathologies. This novel knowledge based on the combination of carbon nanotube imaging and super-resolution methodologies opens the route for the investigation of the physiological and pathological factors leading to different neurodegenerative diseases such as Alzheimer’s, Parkinson’s and Huntington’s diseases.

## 3. Conclusions and Future Outlook

In this review, we discussed the potential of marrying super-resolution techniques with NIR-emitting SWCNTs in the context of several applications. The implementation of SRM for nanomaterials has been relatively limited, mainly due to the low intrinsic PL of individual emitting sites on the nanostructures or the difficulty of labeling them with fluorophores, but recent achievements have demonstrated that SWCNTs have become promising nanostructures in this context. This is due to their intense and photostable NIR PL, which broadly covers the biological and telecom windows (~900–1700 nm); their suitability for single-molecule applications, including for biological samples; and their modularity in shaping exciton emission at defined wavelengths and locations. This modularity opens the route for, e.g., unconventional light emission for data transmission or for spectrally multiplexed imaging in NIR at high resolution. We have described how SRM can generate novel high-resolution images “*in*”, “*of*” and “*with*” SWCNTs. We believe that these achievements lay the foundation for further advancement in the domain of SRM in NIR window and also for the better design and understanding of SWCNT-based structures themselves at the nanoscale. We also believe that the strategies discussed above can guide applications involving other 1D and 2D nanostructures, including semiconducting nanocrystals, h-BN and quantum dots, for instance [89,90,91]. Although many technical and equipment challenges have already been successfully overcome in the quest towards further improvement of spatiotemporal resolution, we foresee more challenges in the near future. These include, for instance, the attainment of 3D super-resolution, which will be the key for better understanding of biological environments [92,93] and complex nanostructures.

## Figures and Tables

**Figure 1 nanomaterials-12-01433-f001:**
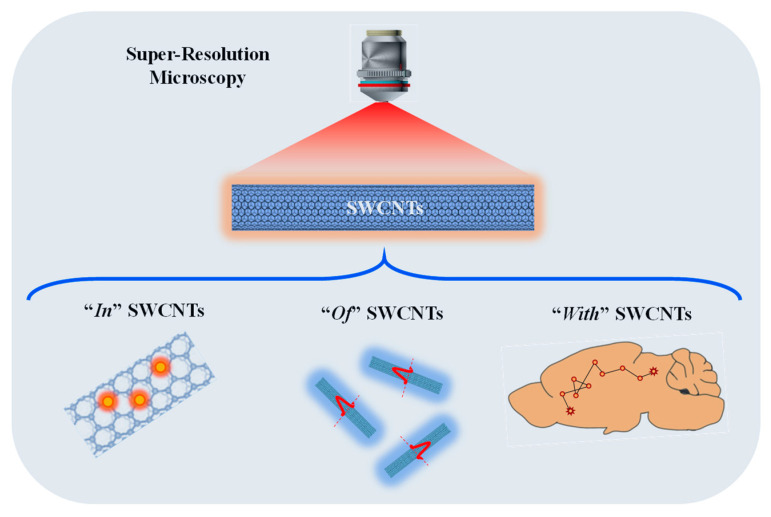
Schematic representation of super-resolution microscopy (SRM) applications involving single-walled carbon nanotubes (SWCNTs).

**Figure 2 nanomaterials-12-01433-f002:**
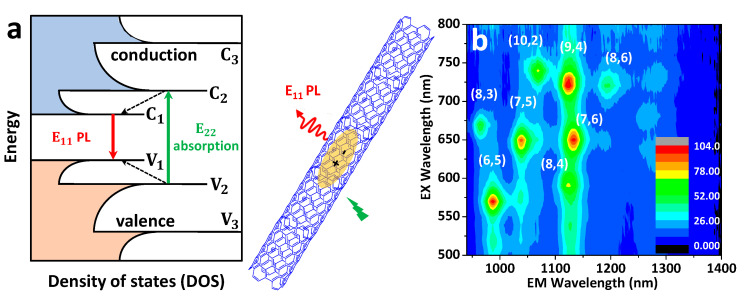
(**a**) A representative scheme of the band-gap structure, depicting plots for the density of states (DOS) with the respective electronic transitions and photophysical properties of single-walled carbon nanotubes (SWCNTs), which lead to the creation of excitons upon excitation at their optical transition energies and subsequent near-infrared (NIR) photoluminescence (E_11_ PL). (**b**) Two-dimensional PL map (excitation–emission profile) of phospholipid-poly(ethylene glycol) (PL-PEG)-functionalized SWCNTs (HiPco). The main nanotube chiralities are indicated.

**Figure 3 nanomaterials-12-01433-f003:**
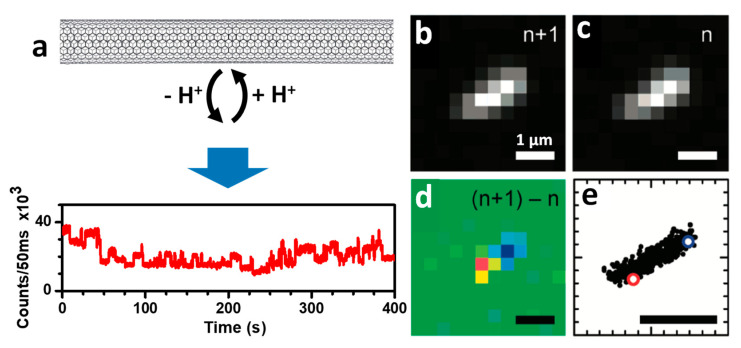
(**a**) Schematic representation of protonation/deprotonation on SWCNTs under acidic condition (pH 4) inducing fluctuation (blinking) of PL intensity. Super-resolution analysis: (**b**–**d**) PL and color-coded differential images; (**e**) the locations of all super-resolved positive and negative intermittency spots. All scale bars are 1 µm. Adapted and modified with permission from [13]. Copyright 2008, American Chemical Society.

**Figure 4 nanomaterials-12-01433-f004:**
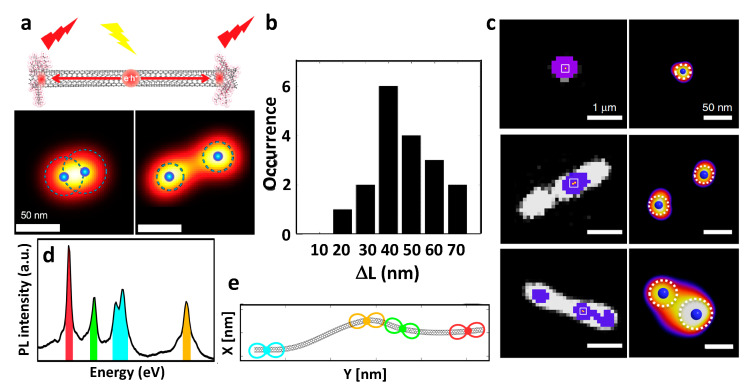
(**a**,**b**) Super-resolved PL images of E_11_* exciton localization in sp^3^ defect-functionalized, fluorescent, ultrashort carbon nanotubes (usCNTs) with the estimated distance (ΔL) between E_11_* exciton locations. The chemically incorporated sp^3^ defects modulate the electronic structure such that the excitons are trapped at the defect sites, leading to bright E_11_* emission. Adapted with permission from [52]. Copyright 2018, American Chemical Society. (**c**) Super-localization of individual sp^3^ defects in long SWCNTs from the diffraction-limited PL images of the defect emission sites (individual defect localization is displayed as blue spots with the corresponding localization precision as dotted circles), reproduced with permission from [61]. Copyright 2019, Springer Nature. (**d**,**e**) Hyperfine spectral feature and super-resolved localization of trapped exciton along the single carbon nanotube axis at cryogenic temperatures, adapted with permission from [62]. Copyright 2019, American Chemical Society.

**Figure 5 nanomaterials-12-01433-f005:**
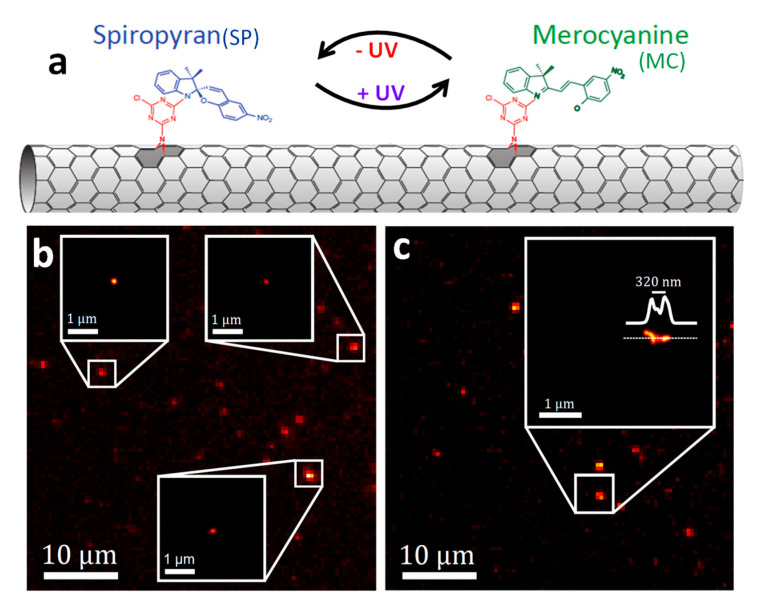
(**a**) Schematic illustration of a photoswitchable hybrid nanomaterial made of (10,2) SWCNTs covalently functionalized with spiropyran-merocyanine (SP-MC). (**b**,**c**) Super-resolved images of individual and closely located SWCNTs, adapted and modified with permission from [64]. Copyright 2019, American Association for the Advancement of Science.

**Figure 6 nanomaterials-12-01433-f006:**
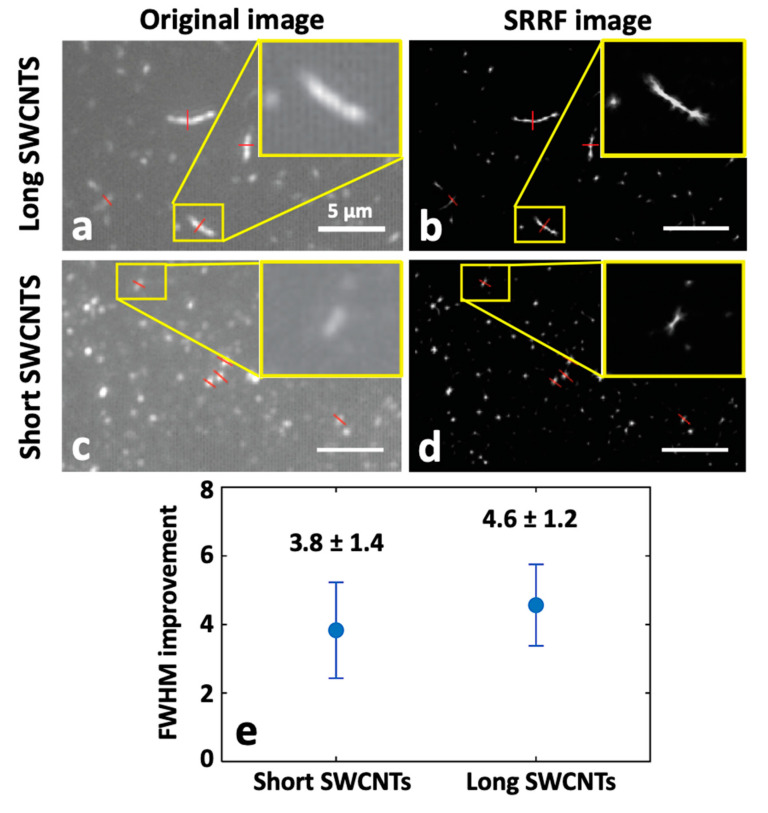
Validation of super-resolution radial fluctuation (SRRF) performance with SWCNTs for (**a**,**b**) long and (**c**,**d**) short nanotubes. (**e**) The super-resolved images upon SRRF analysis with the improvement in the FWHM calculation, adapted and modified with permission from [68]. Copyright 2022, Optical Society of America.

**Figure 7 nanomaterials-12-01433-f007:**
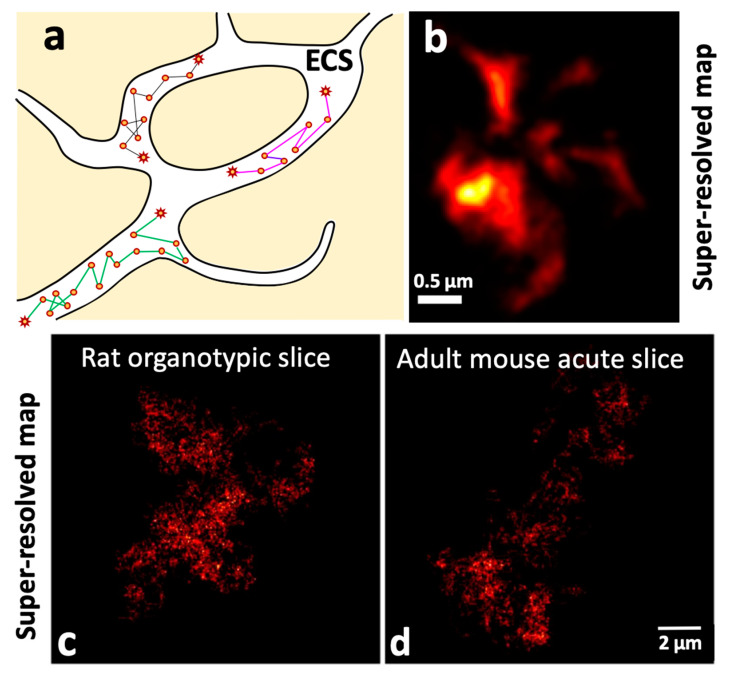
Mapping extracellular space (ECS). (**a**) Schematic depiction of SWCNTs’ diffusion in ECS. Connecting lines in the trajectories do not depict the exact diffusion path but are just for visualizing purposes. (**b**–**d**) Diffusion-analyzed super-resolved images of individual SWCNTs in the ECS of a rat organotypic slice and an adult mouse acute slice; (**b**) reproduced with permission from [83] copyright 2017, Springer Nature and (**c**,**d**) reproduced with permission from [85] copyright 2020, Elsevier.

## Data Availability

Data sharing is not applicable for this review.

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
