# Peer review of "When Super-Resolution Localization Microscopy Meets Carbon Nanotubes"

_nanomaterials, 2022, doi:10.3390/nano12091433_

Round 1

Reviewer 1 Report

In this review by Nandi et al., the authors have discussed the recent advances and development in the characterization of SWCNTs using super-resolution microscopy techniques. The authors present their data with a well-organized figure that, for the most part, is both visually appealing and clear. I think this paper can be Accepted after minor revision. I have several general and specific questions/comments and suggestions, which you can find below:

-From the Introduction, it is not clear why SRM is used for SWCNTs characterization. I think the authors should discuss some recent works in the field and discuss the knowledge gap, and then connect it to the scope of the review.

-An introductory figure presenting all (or at least some of the major) SRM techniques would be very nice for the readers to understand what to expect from the paper.

-Figure 1. All scale bars are of different sizes. Do they represent the same physical size? Generally, it should be of the same length in all panels.

-There should be a separate section discussing the challenges and future outlook in more detail to make the paper attractive to the broad readers of the journal.

Author Response

Reviewer 1

In this review by Nandi et al., the authors have discussed the recent advances and development in the characterization of SWCNTs using super-resolution microscopy techniques. The authors present their data with a well-organized figure that, for the most part, is both visually appealing and clear. I think this paper can be Accepted after minor revision. I have several general and specific questions/comments and suggestions, which you can find below:

Authors’ response: We are thankful to the reviewer for the general appreciation and positive comments on our manuscript. The comments helped us to improve our review article.

  1. From the Introduction, it is not clear why SRM is used for SWCNTs characterization. I think the authors should discuss some recent works in the field and discuss the knowledge gap, and then connect it to the scope of the review.

Authors’ response: We thank the reviewer for suggestion. We have modified the “Introduction” part of the revised manuscript as follows:

Added text in Section 1 (Introduction/Background), line 50 to 69: “For many nanomaterials however, SRM is challenging (…) In this review, we will therefore highlight SWCNTs as a particular class of nanostructures in the context of SRM of SRM.”

  1. An introductory figure presenting all (or at least some of the major) SRM techniques would be very nice for the readers to understand what to expect from the paper.

Authors’ response: We appreciate the reviewer’s suggestion to put an introductory figure and we have added one in the revised version of the manuscript to convey the overall concept of this article.

Note that here, throughout the manuscript we basically focus on the application of SMLM as one of the SRM techniques to SWCNTs and we did not aim at reviewing all SRM techniques that are not applied to nanotubes (we cited them ref 5-9 in the introduction).

Added figure 1.Figure 1. Schematic representation of super-resolution microscopy (SRM) applications involving single-walled carbon nanotubes (SWCNTs).”

  1. Figure 1. All scale bars are of different sizes. Do they represent the same physical size? Generally, it should be of the same length in all panels.

Authors’ response: We thank to the reviewer for pointing out this. We think the reviewer probably means to refer the scale bars for Figure 2 (now 3). We have added one sentence about the scale bar in the caption of Figure 2 (now3) in the revised manuscript.

  1. There should be a separate section discussing the challenges and future outlook in more detail to make the paper attractive to the broad readers of the journal.

Authors’ response: Following reviewer’s suggestion, we have elaborated the “Conclusions and Future Outlooks” section to provide more detail about the conclusion of this article and future directions. Please see the following text added to the conclusion of the revised manuscript:

Added text in Section 3 (Conclusions and Future Outlooks), line 489 to 504:The implementation of SRM to nanomaterials has been relatively limited (…)We also believe that the strategies discussed above will guide applications involving other 1D and 2D nanostructures including semiconducting nanocrystals, h-BN, quantum dots etc [89–91].

Reviewer 2 Report

This review article on the application of super-resolution localization microscopy to single-walled carbon nanotubes (SWCNTs) is timely, interesting, and well-presented. In the manuscript, the authors discussed the recent studies on the super-resolution imaging of NIR-photoluminescent SWCNTs and their applicability for investigating more complex and heterogeneous biological systems. I recommend the publication of this article once the minor issues listed below are taken care of.

  1. Although I know that the authors attempted to focus on the application of super-resolution localization microscopy to SWCNT in the manuscript, the application of this technique to other nanomaterials has been recently reported as well. To cover the broad readership of Nanomaterials, it would be better to include their studies. It has also been described in another recent review paper. (Jeong et al., BKCS, 2022)

  1. The applications of SRM to nanomaterials have been limited, probably due to the difficulty of fluorophore labeling on the nanomaterials and weak photoluminescence from nanomaterials. Such limitations should also be discussed in detail.

  1. The future perspective appears too short as the review article. Please provide more examples of future directions. For example, it may also be interesting if the authors could discuss the possible application of the recently developed spectrally-resolved single-molecule localization microscopic technique (i.e., SR-STORM) to the nanomaterials for spectral imaging because such an approach can provide more information about the nanomaterials. (Zhang et al., Nature Methods, 2015; Kim et al., BKCS, 2021; Jeong et al., BKCS, 2022; Dong et al., Nature Comm, 2016) In fact, the spectroscopic super-resolution imaging of quantum rods and spiropyran-merocyanine isomerization has been recently demonstrated. (Dong et al., ACS Photonics, 2017; Kim et al., JACS, 2017)

  1. The title should be more specified as SWCNT.

Author Response

This review article on the application of super-resolution localization microscopy to single-walled carbon nanotubes (SWCNTs) is timely, interesting, and well-presented. In the manuscript, the authors discussed the recent studies on the super-resolution imaging of NIR-photoluminescent SWCNTs and their applicability for investigating more complex and heterogeneous biological systems. I recommend the publication of this article once the minor issues listed below are taken care of.

Authors’ response: We are thankful to the reviewer for the kind evaluation of our manuscript. We are happy to address all the queries & concerns raised by the reviewer.

  1. Although I know that the authors attempted to focus on the application of super-resolution localization microscopy to SWCNT in the manuscript, the application of this technique to other nanomaterials has been recently reported as well. To cover the broad readership of Nanomaterials, it would be better to include their studies. It has also been described in another recent review paper. (Jeong et al., BKCS, 2022)

Authors’ response: We agree with the reviewer’s comment that it is beneficial to describe the SRM with other nanomaterials for the broad readership. In the “Introduction” section, we had already discussed briefly about other nanomaterials such as gold & silver nanoparticles, 2-D hexagonal boron nitride (h-BN), supramolecular polymers-based nano-assembly, gold nanorods etc.

We have now extended this discussion in the revised manuscript (Section 1, Page 2) as follows:

Added text in Section 1 (Introduction/Background), line 50 to 69: “For many nanomaterials however, SRM is challenging (…) In this review, we will therefore highlight SWCNTs as a particular class of nanostructures in the context of SRM of SRM.”

  1. The applications of SRM to nanomaterials have been limited, probably due to the difficulty of fluorophore labeling on the nanomaterials and weak photoluminescence from nanomaterials. Such limitations should also be discussed in detail.

Authors’ response: Thanks to the referee for this remark. Most of the nanomaterials are indeed not suited for SRM due to either the difficulty of fluorophore labeling on the nanomaterials and weak photoluminescence from nanomaterials or less photostability (photobleaching or blinking).

This is now better discussed in the introduction (see response to point 1)

  1. The future perspective appears too short as the review article. Please provide more examples of future directions. For example, it may also be interesting if the authors could discuss the possible application of the recently developed spectrally-resolved single-molecule localization microscopic technique (i.e., SR-STORM) to the nanomaterials for spectral imaging because such an approach can provide more information about the nanomaterials. (Zhang et al., Nature Methods, 2015; Kim et al., BKCS, 2021; Jeong et al., BKCS, 2022; Dong et al., Nature Comm, 2016) In fact, the spectroscopic super-resolution imaging of quantum rods and spiropyran-merocyanine isomerization has been recently demonstrated. (Dong et al., ACS Photonics, 2017; Kim et al., JACS, 2017)

Authors’ response: As per the reviewer’s suggestion, we have elaborated this section to provide more detail about conclusions of our manuscript and future perspective. Regarding the articles which are suggested by the reviewer, we have added the ones related to nanomaterials (ref 90 and 91):

Added text in Section 3 (Conclusions and Future Outlooks), line 489 to 504: “The implementation of SRM to nanomaterials has been relatively limited (…)We also believe that the strategies discussed above will guide applications involving other 1D and 2D nanostructures including semiconducting nanocrystals, h-BN, quantum dots etc [89–91].

  1. The title should be more specified as SWCNT.

Authors’ response: We thank to the reviewer for the suggestion. However, we want to keep the title of the manuscript concise. Furthermore, in the “abstract” and in the “keywords” of the manuscript, we specifically mentioned that we are mainly talking about SWCNTs. This is why we kept the title unchanged.